# Earthworms act as biochemical reactors to convert labile plant compounds into stabilized soil microbial necromass

Gerrit Angst [1]*, Carsten W. Mueller [2], Isabel Prater[2], Šárka Angst [1], Jan Frouz[1,3], Veronika Jílková[1], Francien Peterse [4] & Klaas G.J. Nierop[4]

Earthworms co-determine whether soil, as the largest terrestrial carbon reservoir, acts as source or sink for photosynthetically fixed $CO_2$. However, conclusive evidence for their role in stabilising or destabilising soil carbon has not been fully established. Here, we demonstrate that earthworms function like biochemical reactors by converting labile plant compounds into microbial necromass in stabilised carbon pools without altering bulk measures, such as the total carbon content. We show that much of this microbial carbon is not associated with mineral surfaces and emphasise the functional importance of particulate organic matter for long-term carbon sequestration. Our findings suggest that while earthworms do not necessarily affect soil organic carbon stocks, they do increase the resilience of soil carbon to natural and anthropogenic disturbances. Our results have implications for climate change mitigation and challenge the assumption that mineral-associated organic matter is the only relevant pool for soil carbon sequestration.

[1] Biology Centre of the Czech Academy of Sciences, Institute of Soil Biology & SoWa Research Infrastructure, České Budějovice, Czech Republic. [2] Chair of Soil Science, Technical University of Munich, Freising, Germany. [3] Faculty of Science, Institute for Environmental Studies, Charles University, Prague, Czech Republic. [4] Department of Earth Sciences and Geolab, Faculty of Geoscience, Utrecht University, Utrecht, The Netherlands. *email: gerrit.angst@bc.cas.cz

S oils represent the largest terrestrial organic carbon (C) reservoir and sustaining its function as long-term sink for photosynthetically fixed $CO_2$ is central to climate change mitigation and sustainable food production. Whether soils act as a C sink or source is largely co-determined by soil fauna, earthworms in particular, as they feed on the organic C that enters the soil, mix mineral soil with organic matter, and substantially affect organic C sequestration[1]. An increase in the abundance of earthworms in ecosystems worldwide through intensified organic fertilisation and the ongoing invasion of North-American forest soils by exotic earthworms[2,3] will further amplify their importance to the terrestrial C cycle.

Earthworms have two principal effects on soil organic C dynamics that work in opposite directions: on the one hand, earthworms increase mineralisation of organic C (i.e., the production of $CO_2$) by stimulating microbial activity, biomass, richness, and diversity[4–9], whereas on the other hand, they stabilise organic C by promoting the formation of macro- and microaggregates[9–11]. The net effect of these adverse processes is still subject to debate[12,13]. Uncertainties arise from the fact that most studies have been conducted on very short timescales, typically not exceeding several weeks (2–7)[2,8,14]. These studies detect a rapid increase in soil organic C mineralisation immediately following the introduction of earthworms to the soil[13,15]. However, a few long-term studies (> 200 days) have shown that the increase in mineralisation of soil organic C might be transient and in fact levels off with prolonged experiment duration[2,16]. In addition, many studies focus on coarse-scale elemental properties, such as C and nitrogen contents of bulk soil and/or density, particle-size or aggregate fractions. Although essential, these measurements do not provide any information on the composition of soil organic matter and the specific response of its different components (such as C derived from plants or microbes) to the presence of earthworms[17–21]. In particular, the exact role of earthworms in the build-up of mineral-protected microbial necromass (i.e., microbial cell walls) in soil aggregates and organo-mineral associations, which recent studies consider crucial to enhance soil C stability[22–24], is unknown[25]. To investigate the long-term effect of earthworms on soil organic C dynamics and stability, we must focus on distinct constituents of soil C.

Here, we combine long-term incubations (33 weeks) of earthworm-affected and non-affected soil with physical fractionation and molecular analyses. We compare bulk, plant and microbial-derived organic C in soil density and particle-size fractions isolated from macro- and microaggregates to show that the presence of earthworms does not necessarily affect soil organic C mineralisation rates or soil organic C contents, but instead causes a substantial molecular alteration of soil organic matter. We reveal that earthworms function as biochemical reactors that stimulate the microbially mediated build-up of soil organic C in aggregates and organo-mineral associations by transferring plant-derived C into more stabilised, microbial necromass. Our findings have substantial implications for long-term organic C sequestration and turnover and modelling of C dynamics in soil.

## Results

**Minor effects on soil organic C mineralisation and contents.** At the end of the 33-week experiment, the presence of earthworms had negligible effects on the mineralisation of soil organic C and bulk soil organic C contents (Figs 1 and 2; $p > 0.1$). To evaluate the effect of earthworms on organic C in more detail, we separated the bulk soil into five fractions increasing in stability: plant fragments freely residing in the soil (fPOM, i.e., a non-protected soil fraction that is most similar to the plant input),

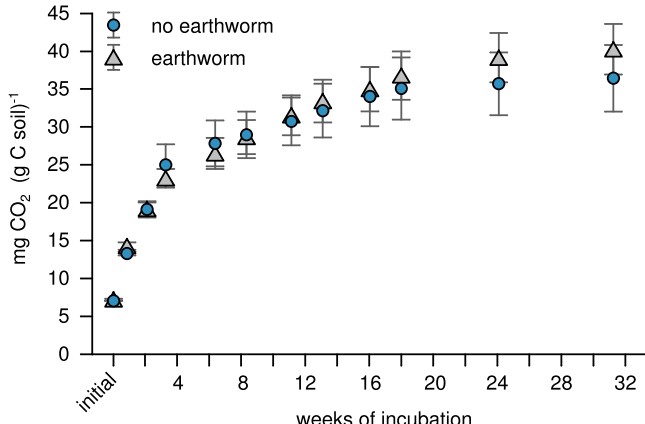

**Fig. 1** Cumulative heterotrophic respiration from earthworm-affected and non-affected soils during the experimental period. These values are indicative of mineralised soil organic C and were calculated by back-titrating NaOH not reacted with evolving $CO_2$ in the closed incubation jars and subsequent normalisation to organic C contents. The figure displays arithmetic means and standard errors from three replicates. Differences between earthworm-affected and non-affected soils were not statistically significant.

particulate organic matter (POM) occluded (i.e., stabilised) within macroaggregates ($oPOM_{macro}$; > 63 μm) and microaggregates ($oPOM_{micro}$; < 63 μm), and organo-mineral associations from macroaggregates ($clay_{macro}$) and microaggregates ($clay_{micro}$). Similar to the bulk soil, organic C contents of these fractions were marginally affected by the presence of earthworms. Yet, we observed lower amounts and C contents of plant fragments freely residing in earthworm-affected soil (fPOM; $p < 0.1$), and higher organic C contents in particulate organic matter occluded in microaggregates ($oPOM_{micro}$; $p < 0.1$; Fig. 2c; with medium to large effect size). Combined, these data indicate a redistribution of earthworm-affected organic matter from less stable (fPOM) to more stable, aggregate occluded soil compartments ($oPOM_{micro}$), with no increase in C mineralisation as reported by many short-term studies[14,26].

**Earthworms alter the composition of soil organic matter.** Evidence for the role of earthworms in altering the molecular composition of soil organic matter is scarce and mainly focused on plant compounds, such as lipids or lignin[17,21,27]. We combined $^{13}C$ nuclear magnetic resonance spectroscopy with the extraction of biomarkers for plant-derived organic matter (hydrolysable hydroxy- and dicarboxylic acids) and microbial necromass (amino sugars) to track both the fate of plant and microbial compounds in soil. Our spectroscopic data indicate a consistent decrease in the relative abundance of lipids and a concurrent increase in the relative abundance of carbohydrates and proteins in earthworm-affected soil fractions (Fig. 3). The only exception to this general pattern was the occluded particulate organic matter from microaggregates ($oPOM_{micro}$), where macromolecule composition was insensitive to earthworm activity (Fig. 3). The spectroscopically determined lipids mainly derive from plants, which is confirmed by an analogous decrease of our plant biomarker, particularly in the less stable soil fractions (up to 83%; fPOM, $oPOM_{macro}$ but also $clay_{macro}$; Fig. 4a). Because we did not add additional plant material during the experiment, parallel increases in the relative amount of carbohydrates and proteins derive from earthworm-excreted mucus that is quickly channelled into soil microbes[28]. This conclusion is, in part, supported by strong increases of microbial necromass in organo-mineral associations from macroaggregates ($clay_{macro}$) and in

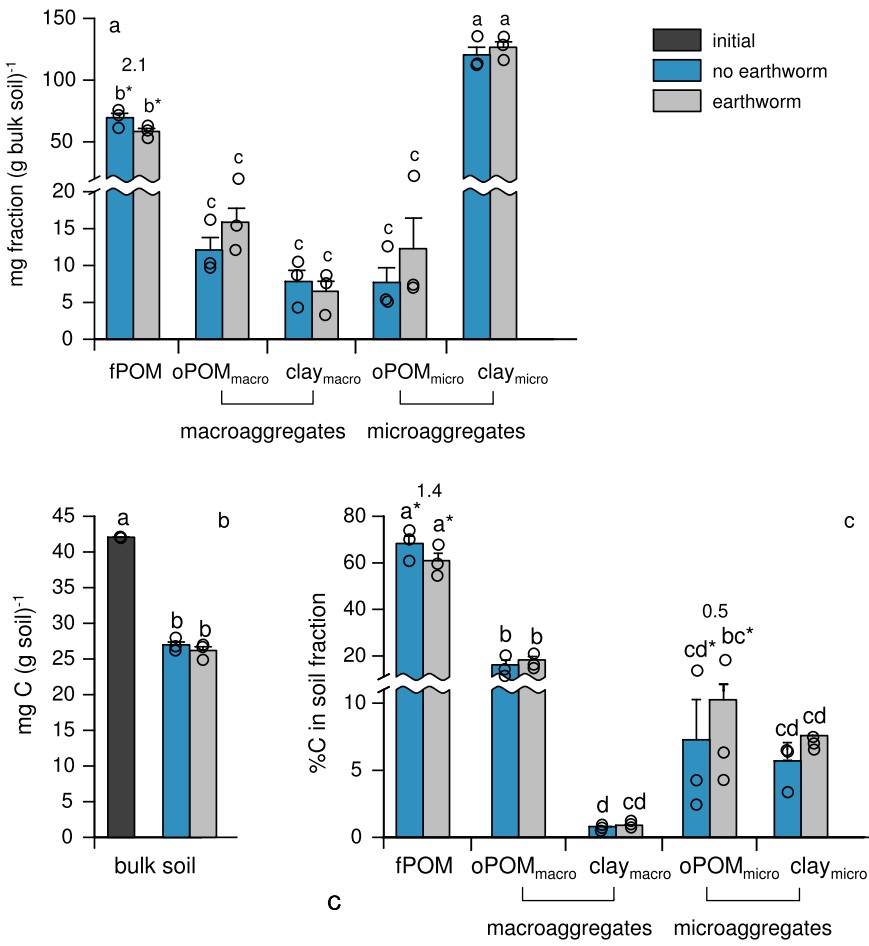

**Fig. 2** Weight of soil fractions **a** and soil organic C in bulk soil **b**, and soil fractions **c** in earthworm-affected and non-affected soils. The soil fractions were separated by a combined density, aggregate and particle-size fractionation. We separated non-protected plant fragments freely residing in the soil and most similar to the plant input (fPOM), particulate organic matter occluded (i.e., stabilised) within macro- ($oPOM_{macro}$) and microaggregates ($oPOM_{micro}$) and organo-mineral associations from macro- ($clay_{macro}$) and microaggregates ($clay_{micro}$). The stability of soil organic matter in these fractions commonly increases in the order fPOM < $oPOM_{macro}$ < $oPOM_{micro}$ < $clay_{macro}$ = $clay_{micro}$. Organic C contents in bulk soil and the soil fractions were measured on an elemental analyser via dry combustion. Significant differences ($p < 0.05$) are indicated by different letters; marginally significant differences ($p < 0.1$) are indicated by asterisks. Error bars indicate standard errors. Results from factorial ANOVA, $n = 15$ (independent treatments), df = 4 **a** and **c**, and $n = 9$ (independent treatments), df = 2 for **b**. Cohen's **d**, as a measure of effect-size, is displayed above values significantly and marginally significantly differing from each other.

$oPOM_{micro}$ in the presence of earthworms (by 37% and 73%, respectively; Fig. 4b), whereas it remained unchanged in the other soil fractions. These only partly matching patterns between microbial necromass and (microbial) protein can be explained by the fact that the spectroscopically estimated protein was calibrated against amino acids[29], such as also occurring in exo-enzymes and other extracellular polymeric substances, and, thus, provides different albeit complementing information with regard to our marker for microbial necromass.

## Discussion

When earthworms feed on organic matter, they add large quantities of easily degradable organic C (mucus) to ingested material[28]. This mucus boosts microbial C use[30,31] and the build-up of microbial biomass in mineral soil[9,11]. These effects are independent of the foraging strategies typical for different earthworm species[32], e.g., litter feeding vs. soil feeding. Our data show that, in this way, earthworms accelerate decomposition of plant-derived organic matter (Figs. 3 and 4a) and the concurrent build-up of microbial necromass (Fig. 4b). Because microbial necromass mainly increases in mineral-associated organic matter and small occluded particulate organic matter, earthworms foster

the sequestration of organic C in more stable, physico-chemically protected C pools (sensu Cotrufo et al.[22])[33].

In the presence of earthworms, the particulate organic matter from microaggregates ($oPOM_{micro}$) shows the proportionately highest increase in microbial necromass (73%; Fig. 4b). This result is surprising given that reactive mineral surfaces (i.a., $clay_{micro}$) have been proposed to be the major binding sites for microbial compounds[34], whereas plant-derived organic matter is still considered to form the organic core of microaggregates (i.e., $oPOM_{micro}$)[35]. Earthworm activity accelerates the turnover of pre-existing aggregates and the mixing of organic debris with microbial material[36], which then act as microbially enriched nuclei for the formation of new microaggregates. Our data suggest that the microbial material in these nuclei is mainly built up from microbial cell walls, whereas microbial material in macroaggregate-occluded particulate organic matter ($oPOM_{macro}$) and organo-mineral associations ($clay_{macro}$ and $clay_{micro}$) is also derived from extracellular polymeric substances, such as exo-enzymes. This inference is based on the fact that the proportion of proteins in earthworm-affected soil increases in all fractions excluding $oPOM_{micro}$ (Fig. 3), in which we record an increase of microbial necromass alone (Fig. 4b). Our study is, to

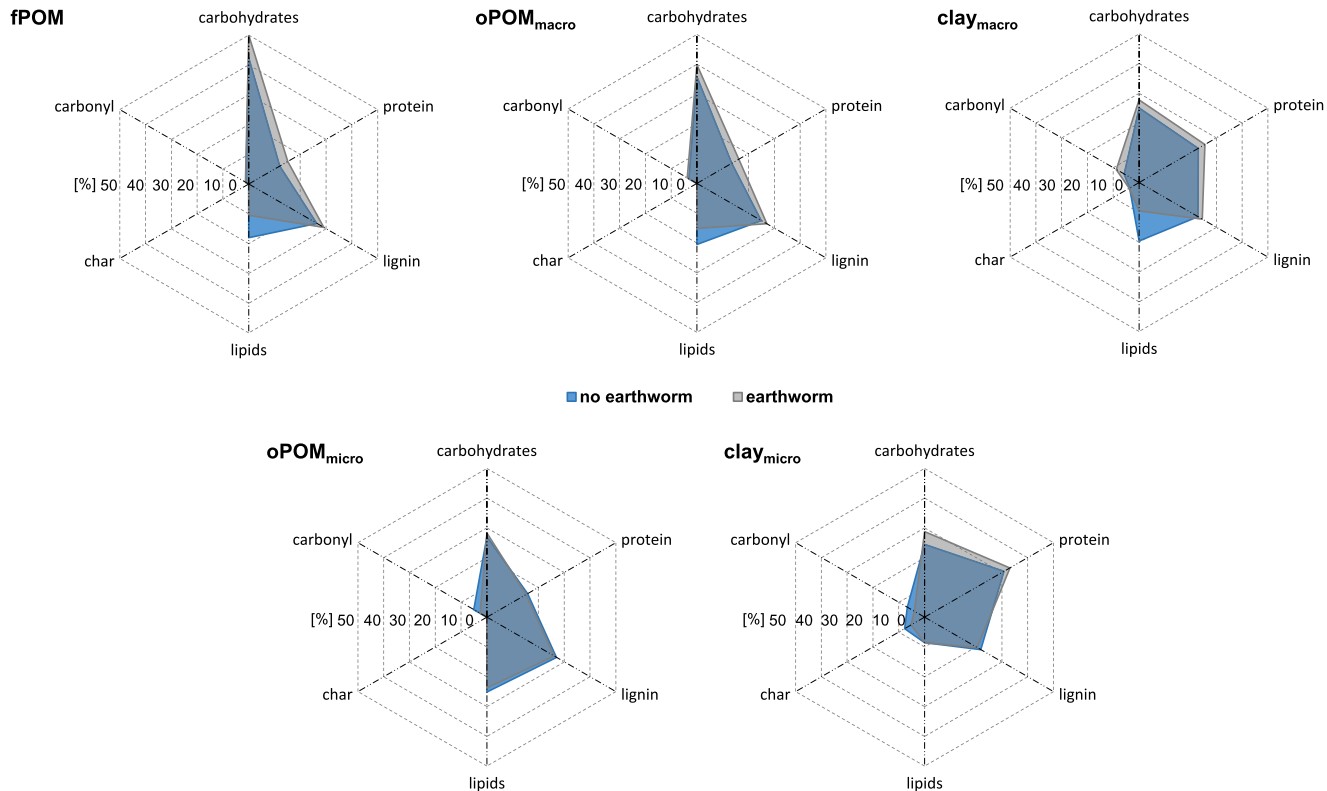

**Fig. 3** Relative contribution of biomolecules (carbohydrates, protein, lignin, lipids, char and carbonyl) to fractions from earthworm-affected and non-affected soils (fPOM, oPOM$_{macro}$, clay$_{macro}$, oPOM$_{micro}$ and clay$_{micro}$). The contribution of biomolecule components was calculated by running a molecular mixing model[29] with data from $^{13}$C nuclear magnetic resonance spectrometry after separating each spectrum into seven integration regions. The model mathematically combines spectral information for compounds of known structure to predict the spectral composition of the sample. The carbohydrates and proteins are derived from earthworm mucus and/or microorganisms and their products because plant material was added only at the beginning of the incubation. Lipids and carbonyl are predominantly plant-derived (see also Fig. 4).

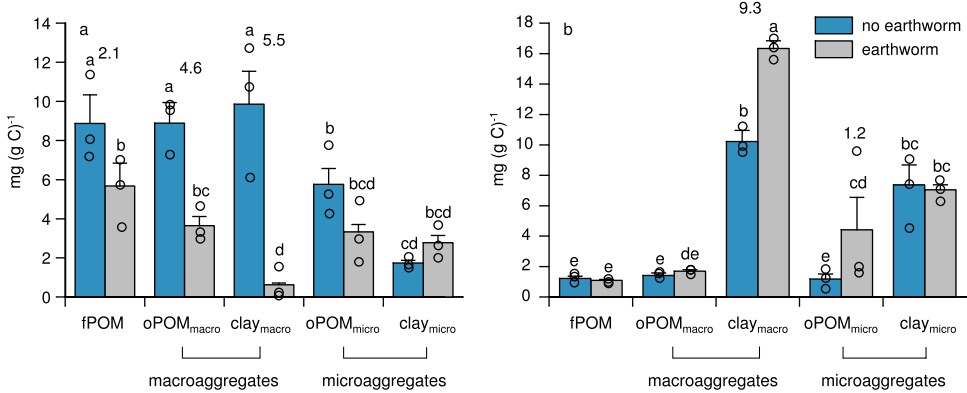

**Fig. 4** Plant **a** and microbial **b** biomarkers in soil fractions from earthworm-affected and non-affected soils. The plant marker was calculated as the sum of dicarboxylic, ω-hydroxy and mid-chain substituted hydroxy alkanoic acids extracted from the soil fractions using a sequential extraction procedure. The microbial marker (indicative for microbial necromass) was calculated as the sum of the amino sugars galactosamine, glucosamine, mannosamine and muramic acid extracted from the soil fractions via acid hydrolysis. Significant differences ($p < 0.05$) are indicated by different letters; marginally significant differences ($p < 0.1$) are indicated by asterisks. Error bars indicate standard errors. Results from factorial ANOVA, $n = 15$ (independent treatments), df = 4. Cohen's $d$, as a measure of effect-size, is displayed above values significantly and marginally significantly differing from each other.

our knowledge, the first to show that microbial necromass can occur in more stable pools of soil organic matter without being associated with mineral surfaces, and increasingly so in the presence of earthworms. In this respect, we highlight the functional importance of small particulate organic matter for C sequestration and challenge the view of mineral-associated organic matter being the only pool focused on when it comes to the stabilisation of C.

Notably, we do not observe a similar increase in microbial necromass in oPOM$_{macro}$ (Fig. 4b), suggesting that earthworms differentially affect the biochemistry of macro- and micro-aggregates. The turnover of macroaggregates is accelerated in the presence of earthworms[37] so that occluded particulate organic matter is continuously released and degraded upon aggregate breakdown. At the same time, earthworms incorporate pre-existing microaggregates into new macroaggregates[38] and

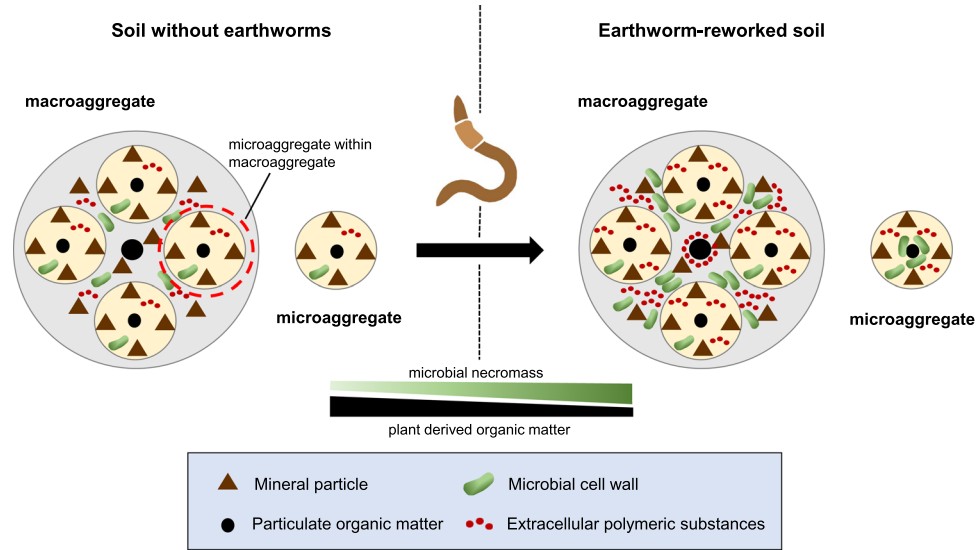

**Fig. 5** Conceptual diagram depicting how earthworms as biochemical reactors affect the molecular composition of differently sized aggregates and soil fractions within. In soil devoid of earthworms, macro- and microaggregates are built up from occluded particulate organic matter surrounded by mineral soil particles. These particles are glued together by microbial material (cell walls and extracellular polymeric substances; EPS). When earthworms rework these soils through burrowing, casting and consumption of organic matter, they change the molecular composition of macro- and microaggregates through their stimulating effect on microorganisms: first, earthworm activity increases the microbial glue (i.e., microbial cell walls and EPS) with which mineral particles and microaggregates cohere in macroaggregates and second, their activity creates nuclei enriched in microbial cell walls for the formation of new microaggregates. These processes are accompanied by an enhanced decomposition of plant-derived organic matter, particularly in the less stable soil fractions (non-protected plant fragments that freely reside in the soil and macroaggregate-occluded particulate organic matter). Ultimately, the presence of earthworms leads to larger amounts of microbial necromass in stable fractions from macro- and microaggregates by which the resilience to disturbances of that C is enhanced.

increase the microbial glue between these structures. Together, these processes explain why microbial necromass remains constant in $oPOM_{macro}$, but increases by 37% in organo-mineral associations ($clay_{macro}$; representing the microbial glue) in these macroaggregates (Fig. 4b). Clearly, earthworms increase the amount of microbial necromass in stable fractions from micro- and macroaggregates.

The enhanced contribution of microbial residues in earthworm-affected and more stable soil fractions ($clay_{macro}$ and $oPOM_{micro}$), though, occurs at the expense of plant compounds, particularly in the less stable fractions (fPOM and $oPOM_{macro}$, but also $clay_{macro}$; Figs. 3 and 4). These adverse processes equilibrate the overall bulk organic C contents and organic C mineralisation between earthworm-affected and non-affected soil (Figs. 1 and 2) and clearly illustrate why the effect of earthworms on soil organic matter dynamics is difficult to reveal by coarse-scale measurements alone. Based on our molecular data, we propose that earthworms function as biochemical reactors (Fig. 5), in which they transform plant-derived organic matter into microbial necromass, and stabilise this necromass in aggregates, without necessarily leaving any major effect on bulk measures, such as the soil organic C content. Our novel concept mechanistically accounts for the discrepancy between the lack of major observable changes in soil organic C stocks in long-term studies[2] and the earthworm commonly referred to as ecosystem engineer, who influences soil organic matter dynamics as a result of burrowing, casting, and consumption of organic matter[1]. Our concept does not necessarily imply that the introduction of earthworms will increase the size of soil organic C stocks, but once established (e.g., in ecosystems previously devoid of earthworms[39]), earthworms will enhance the resilience of organic C in mineral soil against disturbances. The strength of this effect may vary with soil type and, thus, soil properties[40,41], such as sand content to which earthworms are particularly sensitive[42]. However, we expect our concept to be relevant ubiquitously where soil dwelling

earthworms are present, considering their overarchingly positive influence on the microbial biomass in mineral soil[32]. Given that earthworms occur globally in almost every type of ecosystem[43] and generally affect the organic C that is stored in the top ~20 cm of soil, this translates into ~616 Pg organic C[44] that is potentially governed by the proposed mechanism. This mechanism may at least partly compensate for the negative effects of exotic earthworms on forest floor C in invaded ecosystems[45,46] and, in the long-term, climate change mitigation policies would benefit from the maintenance of earthworm populations in mineral soils worldwide. Our findings also emphasise the need to include earthworms in models on C dynamics, which often acknowledge the importance of soil fauna[47,48] but do not explicitly consider it as a parameter. Implementing the functioning of earthworms as biochemical reactors should improve the capability of such models to predict future changes in soil organic C stocks.

## Methods

**Field sampling and incubation experiment**. We complied to some key aspects identified by previous studies[2,13] when choosing the soils for sampling and setting up the incubation experiment (see also below): we chose soil with a low C background not inhabited by earthworms before, we applied earthworm densities comparable to the situation in the field and we ran our incubation for >200 days.

Composite soil samples for the incubation experiment were collected in October 2016 from different locations on the profile walls of a Haplic Fluvisol (B horizon at ~40 cm depth[49], 65% sand, 20% silt, 15% clay, dominated by Illite) under a monocultural alder (*Alnus glutinosa* (L.) Gaertn.) stand (see also Supplementary Table 2). We used subsoil for the experiment to ensure a low C background and no earthworm legacy, but a similar mineralogical/textural composition as compared with the topsoil. Earthworms (*Lumbricus rubellus*, Hoffmeister—an epi-endogeic species—and *Aporrectodea caliginosa*, Savigny, an endogeic species), roots and leaf litter were sampled at the same site from the topsoil or in litter traps, respectively. In the laboratory, soils were dried and passed through a 2 mm sieve. Roots were cleaned from adhering soil by gentle rinsing with deionized water. Leaves and roots were freeze-dried and mechanically crushed to a size of 2 × 2 mm, which facilitates ingestion by earthworms[50]. Directly prior to addition to the incubation treatments, leaves and roots were gently rewetted with deionised water. We established three treatments in five replicates: control (soil without addition of leaves/roots or

earthworms; data on mineralisation of C and C contents in soil fractions in this treatment can be found in Supplementary Fig. 1 and Supplementary Table 1), soil with leaves and roots only, and soil with leaves, roots and earthworms. We added ~80 g of soil to 250 ml glass jars, adjusted the soil to 60% of maximum water holding capacity and mixed 3 g of root material and 4 g of leaf material uniformly with the soil. Using this approach, we pre-empted the mixing of litter with the mineral soil by bioturbation. Yet, this step was needed to define earthworms as the only variable among treatments and obtain a more mechanistic insight into the effects of earthworms on soil organic matter chemistry. After a 7-day pre-incubation period, one specimen of L. rubellus and two specimens of A. caliginosa were added to the earthworm treatments. While establishing the treatments, we payed special attention to choosing earthworms of similar size. The earthworm density (~ 60 individuals m$^{-2}$) and the plant material additions are comparable to the situation in temperate forest soils, considering root to shoot ratios of soil organic matter and consumption rates of organic matter by earthworms[16,51,52]. The glass jars were incubated at 15 °C in the dark for 33 weeks (231 days; ~ corresponding to the period between end and start of litterfall in alder stands[53]) and the water content was regularly adjusted. We could not observe reproductive activity during the experimental period and dead earthworms were directly removed and replaced by a specimen of similar size. After the pre-incubation period of 7 days, we started repeated measurements of heterotrophic respiration. Sodium-hydroxide in small vials not reacted with evolving $CO_2$ after a fixed time in the closed jars was back-titrated with HCl and values normalised to organic C contents to determine the amount (mg) of respired $CO_2$ per g C. After 231 days of incubation, earthworms were removed from the treatments and released back into nature.

**Physical fractionation, spectroscopic, and molecular analyses**. We used a combined density, aggregate, and particle-size fractionation to separate the soils from three replicates per treatment into different organic and mineral soil fractions (a step-by-step description of the fractionation is given elsewhere[54]). In brief, a non-protected, free particulate organic matter fraction (fPOM) was separated with a high-density solution (sodium polytungstate). In a second step, the remaining soil was separated into macro (> 63 µm) and microaggregates (< 63 µm) via wet sieving. Aggregates in both aggregate-size classes were then disrupted using ultrasound to yield occluded (i.e., stabilised) particulate organic matter fractions from macro- (oPOM$_{macro}$) and microaggregates (oPOM$_{micro}$) and the corresponding mineral-associated organic matter fractions (combined fine silt and clay; < 6.3 µm) termed clay$_{macro}$ and clay$_{micro}$. Aggregate stability generally increases from macro- to microaggregates[55]. Values for the density of the heavy liquid, used to separate the particulate from the mineral-associated organic matter fractions, and ultrasonication energies, applied to break up macro- and microaggregates, were chosen to avoid contamination of POM with mineral soil particles and a disruption of POM fractions[56]. To reveal the physical fine scale structure, selected soil physical fractions were imaged using a scanning electron microscope (JEOL JSM-7200F, Freising, Germany; Supplementary Fig. 2).

Carbon and N contents of bulk soil and fractions were determined in duplicate on an elemental analyser (Fisons Instruments NCS NA 1500).

Representative samples of each soil fraction were measured using solid-state $^{13}$C NMR spectroscopy. The samples were spun in a zircon-oxide rotor at 6.8 kHz, with a recycle delay of 0.4 s for the mineral and 1 s for the POM fractions. We accrued at least 4000 scans per sample (several 100.000 for the mineral fractions). Spectra were phase adjusted, baseline corrected, and integrated using the following regions: carbonyl (210–165 ppm), O-aromatic (165–145 ppm), aromatic (145–110 ppm), O$_2$-alkyl (110–95 ppm), O-alkyl (95–60 ppm), N-alkyl/methoxyl (60–45 ppm) and alkyl C (45–10 ppm). We used these integration regions to drive a molecular mixing model, estimating the contribution of various chemical compound classes to each fraction (see below).

Lipids from the soil fractions were sequentially extracted using solvent extraction and base hydrolysis[54]. Methylated and silylated aliquots of the extracts were measured on a GC-MS for lipid identification and on a GC-FID for lipid quantification using squalane as internal standard[54]. We specifically focused on cutin- and suberin-derived hydrolysable lipids, which comprise dicarboxylic, ω-hydroxy, and mid-chain substituted hydroxy alkanoic acids that serve as markers for plant-derived soil organic matter. Lipid concentrations were normalised to the C content of the respective sample and are expressed as mg g C$^{-1}$.

Amino sugars, markers for microbial necromass, were extracted from the soil fractions using a modified protocol by Liang and Balser[57]. In brief, soil fractions were hydrolysed with 6 M hydrochloric acid at 105 °C for 8 h, extracts purified by neutralisation and precipitation of salts in methanol and water and derivatised to aldonitrile acetates. Amino sugars were measured on a GC-FID, quantified by comparison with external standards and their concentrations normalised to the organic C content of the respective soil fraction. Amino sugars are presented as their total sum (i.e., the sum of galactosamine, glucosamine, mannosamine and muramic acid).

**Statistics and calculations**. Statistically significant differences between treatments (earthworm/no earthworm) and soil fractions were tested using repeated measures analysis of variance (ANOVA) for the respiration data and factorial ANOVA for all other data (with treatment and soil fraction as categorical factors; $n = 15$, df = 4). The data were scanned for homoscedasticity and normality and log or square-root

transformed if necessary. Differences are indicated as significant ($p < 0.05$; different letters) and marginally significant ($p < 0.1$; asterisks), following least significant difference post hoc tests. We also calculated Cohen's $d$ for significant and marginally significant differences as a measure of effect-size. Values for $d > 0.8$ indicate large effect sizes, i.e., large effects of treatment (earthworm/no earthworm) on significant and marginally significant differences.

We run the molecular mixing model described by Nelson and Baldock[29] with the NMR data to estimate the relative contribution of six biomolecule components to the individual soil fractions: carbohydrates, protein, lignin, lipids, char and carbonyl. The model mathematically combines spectral information for compounds of known structure to predict the spectral composition of the sample. The model has been shown to account for ≥ 95% of the total NMR signal intensity for terrestrial samples[29].

**Reporting summary**. Further information on research design is available in the Nature Research Reporting Summary linked to this article.

## Data availability
The raw data this manuscript is based on are attached as Supplementary Data 1 and Supplementary Data 2.

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

## Acknowledgements

This study was funded by the Czech Ministry of Education, Youth and Sports (grant number MSM200961705), the Soil and Water Research Infrastructure funded by MEYS CZ (grant numbers LM2015075 and EF16_013/0001782) and the Czech Science Foundation (grant numbers 19-00533Y and 18-24138S). We thank Travis Meador and Roey Angel for discussing and commenting on the manuscript, Katja Heister for facilitating the Geolab infrastructure at the Utrecht University, Thiago Inagaki for help with the collection of SEM images and Sigrid Hiesch, Vladimir Šustr, Kateřina Lapáčková, Tomáš Picek and Jiří Petrásek for help in the laboratory.

## Author contributions

G.A., S.A. and J.F. designed the experiment, G.A. conducted physical fractionations, lipid/amino sugar extractions, performed GC/MS measurements and drafted the manuscript, K.G.J.N. and F.P. supervised the lipid extractions and analyses, I.P. and C.W.M. conducted NMR measurements and V.J. performed CO$_2$ respiration measurements. All authors significantly contributed to manuscript writing and data interpretation.

## Competing interests

The authors declare no competing interests.
