## [Peer Review File · Communications Biology]

Reviewers' comments:

Reviewer #2 (Remarks to the Author):

Summary: This paper gives a very strong and convincing description of exactly what worms do to the soil. The ms elucidates where worms get C and where they end up putting it, and the ms describes the mechanism by which the worms do this. The authors make a big deal about how this helps up understand C sequestration in a global warming scenario, but it also helps us understand what happens to the soil when worm behavior and health is modified from contamination. So its results will be interesting to many soil ecologists and environmental scientists. The writing and critical thinking throughout is strong. The graphics are solid and necessary. Figure 5 is both creative and helpful for reader understanding. I commend the authors on this fabulous article.

Regarding places for improvement, I would like to see photos of the soil fractions to better understand what the worms were doing. If the authors have no such photos, this isn't a deal breaker. Perhaps a schematic would work?

I have no complaint about that stats, although the authors could add a Cohen's d for strength. (It's an easy analysis. Plug in means and standard deviations and get a d. Check out this website for a discussion on what the d adds to statistical understanding and a link to a calculator: http://www.polyu.edu.hk/mm/effectsizefaqs/effect_size_equations2.html)

The authors used bench techniques that I am not familiar with and cannot comment on, but I have several questions about the worms, their origin, their reproduction and size. See my discussion of the Methods below.

In summary, I recommend publishing this paper. It's important, original and solid. As far as I know, no one else has addressed this issue. Before publication, the authors should answer the questions about the worms (see below), and consider images of the soil fractions and consider Cohen's d. There are small places for improvement described in the sections below.

Abstract: Solid and clear writing. I understand what the authors are trying to say and share their enthusiasm for what their findings represent. Two small typos. On line 18, the comma goes inside the quotation mark. I don't think you need to put quotation marks around reactors on line 19.

Introduction: I think you can delete the 'and' in line 31. I love the opening paragraph. I'd reference this paper on the opening alone. Good citations. I don't like the use of the work bulk on line 46 and then again so close on lines 47 and 48. The writing is so solid that this detracts from it.

This is a powerful introduction, setting the groundwork for an interesting experiment.

Results 1: Negligible effects of earthworms on soil organic C mineralisation and organic C contents in bulk soil and differently stable fractions. Is there any way you could graphically or photographically communicate what the five fractions of bulk soil look like? It would make this section more intuitive to the reader. Line 79: lower weights of what? Carbon? Please clarify. The rest is excellent. Clear and convincing findings. Solid writing. Interesting experiment.

Figures 1 and 2 are solid and professional.

Result 2: Earthworms substantially alter the molecular composition of soil organic matter. I have no substantive comments. The writing is clear and concise. The argument is sound and convincing.

Figures 3 and 4 are clear and concise and add to the understandability of the argument. I'm not sure why the authors mention asterisks since they didn't use them.

Result 3: Earthworms as "biochemical reactors." Again, good writing and good argumentation. Figure 5 adds to the clarity of the findings and makes the results easier to understand. Maybe EPS should be mentioned in the general text.

Methods

Given the length of this study, I'm surprised the worms didn't reproduce. (Mine do.) Did they? Did they reproduce equally across treatments? Did you weigh the worms before and after the project began? You state that when the worms died, you added new ones, but could you tell when the worms died? Even in your small jars, I can't imagine that you could see all the worms all the time and they decompose so quickly. (Less than 24 hours for mine.) Differences in worm biomass could drive differences between jars, which might explain the high standard error of the mean in figure 2b. Where did the worms come from? These questions need to be addressed before publication.

I'm not familiar enough with the physical fractionation, spectroscopic and molecular analyses to comment on those methods. Soil respiration technique is solid.

The stats are fine. The authors might consider using a Cohen's d instead of post hoc comparisons, but it's not a deal breaker. They could even consider use both a Cohen's d in addition to the post hoc tests. This lets the readers know how many standard deviations separate two measures and gives the reader an idea of how big the actual difference is.

In the supplementary material, I sure wouldn't mind seeing the raw data in tabular form. I'd really like to see how similar the data was in each treatment, raw form.

Best,
Sharon T. Pochron, PhD
Sustainability Studies
School of Marine and Atmospheric Sciences
Stony Brook University

Reviewer #3 (Remarks to the Author):

The authors in this work investigate the role of earthworms in the process of soil C stabilizing or destabilizing in frame of the long-term experiment. They found out that the earthworms support the conversion of plant-derived compounds from less stabilized C pools to microbial necromass in more stabilized C pools. In addition, their study is the first to show that substantial portion of this stabilized microbial necromass C is not associated with mineral surfaces, and thus they confirmed significance of particulate organic matter for long-term C sequestration. The authors also pointed out that earthworms do not necessarily affect the soil organic C stock, but affect the form of C in soil. The authors emphasize the importance of these findings for climate change strategies obtained from the long-term experiment that are inconsistent mainly with the results of other authors obtained from short-term experiments.

The paper is well structured and written. It presents the new original and valuable results achieved by modern and high-quality methods and analyses. The conclusions are supported by the appropriate statistical analysis. Because currently little is known about the specific role of earthworm in C cycle

from quantitative (C stock) and mainly qualitative (C transformation processes leading to stabilizing or destabilizing C) point of view, the results can contribute into a mosaic of findings leading to clarification of the irreplaceable role of earthworms in soil ecosystem.

Despite the high value of the results achieved, it is not appropriate to generalize them for soil. It is necessary to keep in mind the high heterogeneity of soil. And the experiment simulated the specific conditions. I recommend to add such explanation for readers.

Despite the fact that experiments of this type are very demanding on maintenance and analysis more variants can contribute to more objective results (more soil types, different soil texture, different earthworm species, etc.). For example percentage of sand particles in soil has big impact on earthworms. Earthworms are quite sensitive to the sharp sand surface and often prefer environment with higher portion of clay particles and thus more clay soils. All these factors can influence C transforming processes and the earthworm role. The setting of factors at the beginning of the experiment has big impact on reached results.

I would also appreciate in the paper (not only as reference) more basic information concerning to the tested soil, at least basic soil chemical (pH, C:N, C:P, nutrients content) and physical (bulk density, porosity, etc.) properties in simple table form of both soil horizons (A and B) at the beginning of the experiment and at the end (of used horizon). Maybe using of both horizons in the experiment, not only B, would show interesting results and comparison.

Overall, the work is valuable and innovative and comments can serve as inspiration for further research. More research work in this field is necessary for formulating of more objective conclusions that can be adopted as appropriate measures in soil management not only leading to climate change but also other soil threats mitigation.

Specific comment:

Page 4, Fig. 1, Figure caption row. 3 – use lower index in CO₂

Reviewers' comments:

Reviewer #2 (Remarks to the Author):

Summary: This paper gives a very strong and convincing description of exactly what worms do to the soil. The ms elucidates where worms get C and where they end up putting it, and the ms describes the mechanism by which the worms do this. The authors make a big deal about how this helps up understand C sequestration in a global warming scenario, but it also helps us understand what happens to the soil when worm behavior and health is modified from contamination. So its results will be interesting to many soil ecologists and environmental scientists. The writing and critical thinking throughout is strong. The graphics are solid and necessary. Figure 5 is both creative and helpful for reader understanding. I commend the authors on this fabulous article.

We sincerely thank Dr. Pochron for commending our work and appreciate the time she invested into reviewing the manuscript.

Regarding places for improvement, I would like to see photos of the soil fractions to better understand what the worms were doing. If the authors have no such photos, this isn't a deal breaker. Perhaps a schematic would work?

We booked some time at the SEM facility of the Technical University of Munich and took SEM images of selected soil fractions. These were attached as supplementary figure 2. These images provide an overview of the fractions separated and investigated in this study.

I have no complaint about that stats, although the authors could add a Cohen's d for strength. (It's an easy analysis. Plug in means and standard deviations and get a d. Check out this website for a discussion on what the d adds to statistical understanding and a link to a calculator: http://www.polyu.edu.hk/mm/sizeeffectsizefaq/effect_size_equations2.html)

We calculated Cohen's d for all significant (and marginally significant) differences and added those values to the respective figures. We also added text to the figure captions explaining the values and a short description to the methods section (L. 268 - 271).

The authors used bench techniques that I am not familiar with and cannot comment on, but I have several questions about the worms, their origin, their reproduction and size. See my discussion of the Methods below.

In summary, I recommend publishing this paper. It's important, original and solid. As far as I know, no one else has addressed this issue. Before publication, the authors should answer the questions about the worms (see below), and consider images of the soil fractions and consider Cohen's d. There are small places for improvement described in the sections below.

Abstract: Solid and clear writing. I understand what the authors are trying to say and share their enthusiasm for what their findings represent. Two small typos. On line 18, the comma goes inside

the quotation mark. I don't think you need to put quotation marks around reactors on line 19.

We erased the quotation marks in line 19 as suggested. However, we could not find any comma in the sentence ending in line 18. We, thus, did not change anything here.

Introduction: I think you can delete the 'and' in line 31. I love the opening paragraph. I'd reference this paper on the opening alone. Good citations. I don't like the use of the work bulk on line 46 and then again so close on lines 47 and 48. The writing is so solid that this detracts from it.

We agree that the use of "bulk" might have potential for misunderstanding. Please note that the word "bulk" is intended to describe measures that do not go into much detail (unlike molecular level analyses). However, we exchanged the word in L. 46 with "coarse-scale" and in L. 48 with "these", but maintained it in L. 47 because it is a common term in soil science to refer to the total soil.

This is a powerful introduction, setting the groundwork for an interesting experiment.

Thank you for the compliment.

Results 1: Negligible effects of earthworms on soil organic C mineralisation and organic C contents in bulk soil and differently stable fractions. Is there any way you could graphically or photographically communicate what the five fractions of bulk soil look like? It would make this section more intuitive to the reader. Line 79: lower weights of what? Carbon? Please clarify. The rest is excellent. Clear and convincing findings. Solid writing. Interesting experiment.

Thanks for the compliments. As noted above, we attached SEM images of the fractions in supplementary figure 2. In (now) L. 78 we refer to the weight of the fPOM fraction. We exchanged "weight" with "amount", so the sentence may be clearer now.

Figures 1 and 2 are solid and professional.

Thank you again for the compliments.

Result 2: Earthworms substantially alter the molecular composition of soil organic matter. I have no substantive comments. The writing is clear and concise. The argument is sound and convincing. Figures 3 and 4 are clear and concise and add to the understandability of the argument. I'm not sure why the authors mention asterisks since they didn't use them.

We are happy about your very positive feedback and thank for pointing out the unnecessary description of asterisks in the caption to figure 4. We deleted it.

Result 3: Earthworms as "biochemical reactors." Again, good writing and good argumentation. Figure 5 adds to the clarity of the findings and makes the results easier to understand. Maybe EPS should be mentioned in the general text.

We are very pleased to get a compliment on the writing as non-native speakers. We added “(EPS)” in the text (L. 107), so readers are familiar with the term when reading the figure captions.

Methods

Given the length of this study, I’m surprised the worms didn’t reproduce. (Mine do.) Did they? Did they reproduce equally across treatments? Did you weigh the worms before and after the project began?

We checked the treatments daily and could not observe any reproductive activity of the earthworms. We added a short statement on this to the methods section (L. 214/215). We did not weigh the earthworms but we payed special attention to choose earthworms of similar size when establishing the treatments. We added a statement to the methods section (L. 208/209).

You state that when the worms died, you added new ones, but could you tell when the worms died? Even in your small jars, I can’t imagine that you could see all the worms all the time and they decompose so quickly. (Less than 24 hours for mine.) Differences in worm biomass could drive differences between jars, which might explain the high standard error of the mean in figure 2b.

As stated above, we checked the treatments daily and did our best to remove dead earthworms as quickly as possible. In some cases, decomposition had already begun but the shape of the earthworms was still clearly visible. We believe the relatively high SE in Figure (we believe the reviewer refers to 2c), cannot be due to decomposing earthworms because the other fractions should have been similarly affected. Yet, standard errors are lower for these fractions. Furthermore, it is more likely that any decomposing earthworm has a more severe effect on the clay fractions because, as far as we know, decomposing earthworms do not accumulate as particulate organic matter in soil, which oPOMmicro (the fraction with the comparably high SE) represents.

Where did the worms come from? These questions need to be addressed before publication.

The worms came from the topsoil overlying the subsoil used for the incubation. This is described in L. 191 – 193. We payed particular attention to the use of soil devoid of earthworms for our experiment.

I’m not familiar enough with the physical fractionation, spectroscopic and molecular analyses to comment on those methods. Soil respiration technique is solid.

The stats are fine. The authors might consider using a Cohen’s d instead of post hoc comparisons, but it’s not a deal breaker. They could even consider use both a Cohen’s d in addition to the post hoc tests. This lets the readers know how many standard deviations separate two measures and gives the reader an idea of how big the actual difference is.

We followed the reviewer’s suggestion, calculated Cohen’s d and added these values to the significant and marginally significant differences displayed in the figures. We also changed the

figure captions and methods section accordingly (please also refer to the answer of a previous comment).

In the supplementary material, I sure wouldn't mind seeing the raw data in tabular form. I'd really like to see how similar the data was in each treatment, raw form.

Please note that the raw data will be published with Dryad if the manuscript will be accepted for publication. Anyway, we also attached the raw data as supplementary table 3.

Best,
Sharon T. Pochron, PhD
Sustainability Studies
School of Marine and Atmospheric Sciences
Stony Brook University

Reviewer #3 (Remarks to the Author):

The authors in this work investigate the role of earthworms in the process of soil C stabilizing or destabilizing in frame of the long-term experiment. They found out that the earthworms support the conversion of plant-derived compounds from less stabilized C pools to microbial necromass in more stabilized C pools. In addition, their study is the first to show that substantial portion of this stabilized microbial necromass C is not associated with mineral surfaces, and thus they confirmed significance of particulate organic matter for long-term C sequestration. The authors also pointed out that earthworms do not necessarily affect the soil organic C stock, but affect the form of C in soil. The authors emphasize the importance of these findings for climate change strategies obtained from the long-term experiment that are inconsistent mainly with the results of other authors obtained from short-term experiments.

The paper is well structured and written. It presents the new original and valuable results achieved by modern and high-quality methods and analyses. The conclusions are supported by the appropriate statistical analysis. Because currently little is known about the specific role of earthworm in C cycle from quantitative (C stock) and mainly qualitative (C transformation processes leading to stabilizing or destabilizing C) point of view, the results can contribute into a mosaic of findings leading to clarification of the irreplaceable role of earthworms in soil ecosystem.

We thank the reviewer for the time invested in the review of our manuscript and the favourable evaluation.

Despite the high value of the results achieved, it is not appropriate to generalize them for soil. It is necessary to keep in mind the high heterogeneity of soil. And the experiment simulated the specific conditions. I recommend to add such explanation for readers.

Despite the fact that experiments of this type are very demanding on maintenance and analysis

more variants can contribute to more objective results (more soil types, different soil texture, different earthworm species, etc.). For example percentage of sand particles in soil has big impact on earthworms. Earthworms are quite sensitive to the sharp sand surface and often prefer environment with higher portion of clay particles and thus more clay soils. All these factors can influence C transforming processes and the earthworm role. The setting of factors at the beginning of the experiment has big impact on reached results.

We thank the reviewer for this comment, which better helps putting our results into perspective. We intended to use widely distributed soil types and textures and thus ended up with the presented approach and material. We revised our statement on the transferability of our results in the discussion accordingly (L.166 - 168)

I would also appreciate in the paper (not only as reference) more basic information concerning to the tested soil, at least basic soil chemical (pH, CHA:CFA, nutrients content) and physical (bulk density, porosity, etc.) properties in simple table form of both soil horizons (A and B) at the beginning of the experiment and at the end (of used horizon). Maybe using of both horizons in the experiment, not only B, would show interesting results and comparison.

We appreciate the detail-oriented attitude of the reviewer and added the requested information for the initially used B horizon in the jars to the manuscript in the form of a supplementary table (pH, N content, bulk density). However, because we only measured these properties for the initial soil and neither measured porosity nor the ratio of humic versus fulvic acids (which is based on a concept increasingly doubted; please refer to Lehmann and Kleber, 2015, Nature), we can, unfortunately, not provide this information. Yet, we believe this is not needed to have a sufficient context for the study, specifically because we performed a thorough physical fractionation of the soil (as with respect to CHA:CFA). We agree that the investigation of different horizons might bring more specific insights and we will consider this comment in future experiments.

Overall, the work is valuable and innovative and comments can serve as inspiration for further research. More research work in this field is necessary for formulating of more objective conclusions that can be adopted as appropriate measures in soil management not only leading to climate change but also other soil threats mitigation.

Specific comment:

Page 4, Fig. 1, Figure caption row. 3 – use lower index in CO₂

Changed as suggested

REVIEWERS' COMMENTS:

Reviewer #2 (Remarks to the Author):

Of all the papers I've read this year, the manuscript by Angst et al. is the most stunning. Not only do the authors clear up one of the biggest mysteries surrounding worms, soil and microbe--do worms turn soils into CO2 sources or sinks--but they also lay out, piece by pieces, the ecosystem services provided by earthworms and their interrelatedness with microbes. This is a stunning piece of scholarship that I will be referencing in all of my upcoming work.

Regarding the revisions, the authors clarified their statistics and the status of worm reproduction, which were the only major edits I'd suggested. The authors also addressed the minor issues the other reviewer and I had brought up, too.

In my opinion, this manuscript is ready for publication. Congratulations on work well done.

Reviewer #3 (Remarks to the Author):

As I saw the revised manuscript I can say that the authors taken into account all my comments and I agree with publication of the article.